# Multi-Point Time-Synchronized Waveform Recording for the Analysis of Wide-Area Harmonic Propagation

**Stanislav Babaev** [1,*] **, Ravi Shankar Singh** [1,*] **, Sjef Cobben** [1] **, Vladimir Ćuk** [1] **and Allan Downie** [2]

[1] Electrical Energy Systems Group, Eindhoven University of Technology, 5600 MB Eindhoven, The Netherlands; j.f.g.cobben@tue.nl (S.C.); v.cuk@tue.nl (V.Ć.)

[2] Power Networks Demonstration Centre, University of Strathclyde, Cumbernauld G68 0EF, UK; allan.s.downie@strath.ac.uk

* Correspondence: S.Babaev@tue.nl (S.B.); r.singh1@tue.nl (R.S.S.)

**Abstract:** This paper focuses on studying the phenomenon of harmonic distortion propagation through distribution networks. This phenomenon is governed by a combination of factors involving the nature of harmonic loads and their dynamic interaction, the influence of background voltage distortion, and harmonic impedance values. The objective of the proposed research includes evaluation of the network response at different nodes to harmonic current injections via utilizing a time-synchronized distributed measurement system. The study is performed in a fully controlled and flexible test network with three medium voltage/low voltage (MV/LV) distribution substations and several managed LV harmonic sources, namely PV inverter, single-phase EV charger and emulated harmonic load with reference current injections. A selection of the results is analyzed and interpretation of the observed phenomena is given with implications that synchronized harmonic measurements can be considered as potential powerful instruments for analyzing power quality disturbances.

**Keywords:** power system harmonics; measurement techniques; power quality; Phasor Measurement Unit (PMU) application in distribution grid

## 1. Introduction

Harmonic emission evaluation has been known as complex problem for many decades. Fluctuating nature of harmonics, high level of diversity and additional implications which appear with interconnection of Renewable Generation (RG) require improvement of the existing assessment and measurement methods. The problem is determination of the origin of harmonic distortion in the case when multiple sources of harmonics are connected to the network.

The system-wide studies of harmonic propagation phenomena and emission assessment present challenges for researchers and engineers. In essence, the broad term 'propagation' includes several specific harmonic-related mechanisms among which are:

- Harmonic interaction
- Phase angle diversity and cancellation
- Harmonic attenuation
- Impact of system and load impedance

Given the dynamic nature of the mentioned processes an appropriate solution for investigation of harmonic propagation in specific parts of the power grid would be the application of multi-point distributed measurement system synchronized to a precision clock with high accuracy. Such a

measurement system configuration will allow the study of the harmonic response of a system to the current injection from diverse loads at different points of the electrical network.

For power system analysis (not specifically PQ) synchronized measurements have already been proposed as a method for resolving several engineering tasks such as stability assessment, fault location, distribution state estimation, and event detection [1–3]. For these purposes, PMU-based infrastructure proved to be an efficient solution. Synchronized phasors reported by each PMU must comply with synchronization accuracy requirement of not worse than ±31.7 μs which is satisfactory for most of the applications sought by power system engineers.

On the other hand, as authors of [4] have shown in their laboratory experiment, much stricter time synchronization requirement of not worse than ±1 μs has to be imposed when a fundamental goal is to process harmonic spectral components.

Given the sub-microsecond time synchronization accuracy, the authors of [5] investigated the feasibility of assigning harmonic responsibilities based on data provided by distributed measurement system installed in small test bench power network. In [6], a methodology for calculating synchronized harmonic phasors was proposed. This methodology aimed to accurately resolve harmonic phase angles while reallocating samples between two subsequent PPS edges. This allows considerations to be taken with regard to deviations in fundamental frequency.

Some studies investigated methodologies on determining individual harmonic current contributions. In [7], the authors reported results of an experiment involving synchronized measurements at large industrial plant facilities. Furthermore, in [8], by utilizing GPS synchronized PQ recorders installed as part of a real high voltage (HV) network, the authors investigated the phenomena of dynamic interaction of harmonic sources with special attention given to the grid code compliance.

Large-scale harmonic measurements with provision of time-synchronized instruments were reported in [9]. The experiment was performed in a distribution system characterized by high penetration of RG. Values of harmonic amplitudes and intersite phase angles have been recorded and interpretation of results has been provided in comparison with the outcome of computer modeling. It is worth noting that PQ measurement techniques based on synchronized sampling are currently being developed not only for distortion-related issues but also for other PQ events, for instance dips, swells, and interruptions. An example of such a study was presented in [10] with a focus on offshore wind farms [4].

As current state-of-the-art shows that new measurement methods are vital for resolving challenges which are brought by the operation of modern power systems. These methods, depending on the PQ phenomena in question have to be linked to specific data processing algorithms, as requirements for these can vary substantially with respect to the final goal.

Still, the challenge in implementing such measurement systems for distortion evaluation lies to a large extent in the lack of commercially available instrumentation and the cost associated with customized solutions. On the other hand, as it was discussed above, stricter requirements on synchronization accuracy and specific signal processing algorithms (particularly for calculating harmonic phase angles) would be realized on these devices had these guidelines been reflected in the responsible standards, which is not the case at the current moment.

In this paper, we report the results of a large-scale experiment performed at a self-contained power network which emulates a real distribution grid operating at MV and LV levels. In this network, a set of synchronized waveform recorders is available at different points. The scope of this work is to investigate transfer of harmonic distortion between MV and LV sides. The test setup is characterized by high degree of flexibility and full control over experiments. The harmonic impact of solar power inverter and electrical vehicle charger is assessed in this work. Additionally, a synthetic harmonic current profile was generated as reference injection.

The paper is organized as follows: in Section 2, the motivation for this study is given and objectives are explained, Section 3 focuses on the description of test setup, utilized measurement system together with the explanation of executed test cases. Additionally, the methodology for harmonic phasor

estimation is briefly discussed. The analysis and interpretation of results is presented in Section 4 and conclusions are given in Section 5.

## 2. Objectives and Motivation

The core of the project is employment of multi-point distributed measurement system synchronized to the precision clock with high accuracy. This measurement configuration allows the evaluation of power network response at different nodes to time-varying harmonic current injections. The research project advances the knowledge in the field of power quality and introduces new approaches and measurement techniques for assessment of harmonic distortion and its impact on wide networks. Additionally, the results of this study are expected to further facilitate integration of devices with global synchronization of the samples and their utilization for power quality assessment.

The objectives of the research comprise investigation of the phenomena of harmonic interaction between background voltage harmonic distortion and harmonic current injections of the nonlinear sources under the study [11–13]. Furthermore, with provision of experimental measurements a diversity of harmonic sources is evaluated. The diversity of phase angles of different harmonic-producing sources is what leads to cancellation effects between harmonic currents [14,15]. According to the IEC 61,000 family of standards, these effects are accounted by applying summation exponents to magnitudes of harmonic currents. To improve the evaluation of harmonic current components, a grid measurement technique which relies on time synchronization is necessary. This takes into consideration time variation of current components and preserves current phase angles. The latter factor results in vectorial summation of currents emitted by different harmonic sources. At this stage, the accuracy of the measurement instrumentation and time synchronization are vital for accurate evaluation of harmonic emission. This type of analysis and configuration of measurement system provides an advantage over conventional power quality assessment methods. Therefore, a voltage harmonic distortion can be correlated in time with harmonic current injections at different locations.

Moreover, an additional objective is to investigate the influence of network impedance on voltage distortion levels. The outcome of a harmonic analysis can differ given the situation when absolutely identical harmonic producing loads operating at identical power levels are connected to the system with dynamically changing impedance. The spectral components extracted out of synchronized data are anticipated to provide a good insight into the consequences of varying system impedance.

## 3. Measurement Campaign

### 3.1. Description of the Test Setup

The Power Networks Demonstration Centre (PNDC) located in Cumbernauld, Scotland is a multifunctional power engineering facility. Among its purposes is a realistic technology testing and execution of the research projects aimed to facilitate development of smart grid sector. PNDC is in possession of a large power network, with a substantial part of it having unprecedented levels of flexibility and control over the real-time operation.

For the purpose of harmonic propagation study, a part of the test network was configured. The diagram of this power network is shown in Figure 1. All in all three MV/LV distribution substations are connected to the main MV busbar (primary switchboard) via different impedance sections representing underground MV cables. There is a possibility to switch between sections A and B when energizing Substation G. Substation D on the other hand can be connected on demand by means of two circuit breakers at both ends of impedance MI 2.

The setup offers a choice between two distinct power sources: the network can be energized by means of 11 kV public grid directly connected to Site Services Switchboard or alternatively via a motor-generator (MG) group providing stable 50 Hz output containing only fundamental frequency component. By engaging this power source, it is possible to provide a non-distorted voltage at Primary Switchboard. At this fashion, we could make sure that fluctuating MV background voltage

harmonics do not enter the setup after MG-set and minimum deviations of fundamental frequency could be secured.

Though, as it was explained above, while background voltage harmonics are not to enter the test network from the upstream MV grid when MG-set is employed, another source of voltage distortion was present in this network. The distortion originated from the series of pole-mounted MV/LV transformers connected to the MV busbar of Substation G. These transformers produced magnetizing current while being energized via impedance section A. Alternatively, when bypassing connection via impedance section B, the primary windings of the pole-mounted transformers become disconnected from the rest of the network. The impact of this harmonic distortion will be mentioned later in corresponding test cases.

Several specific loads were configured and used for this study:

1.　3-phase 10 kVA solar inverter SMA Sunny Tripower
2.　EV charger with Nissan Leaf car capable of operating at 1-phase AC 6 kW
3.　Triphase in current source mode with programmed industrial harmonic profile (30 A fundamental current)
4.　Banks of linear loads operating at unity power factor

The control of the most of the devices can be performed through SCADA-software; however, in some cases, manual switching actions and subsequent check procedures were required.

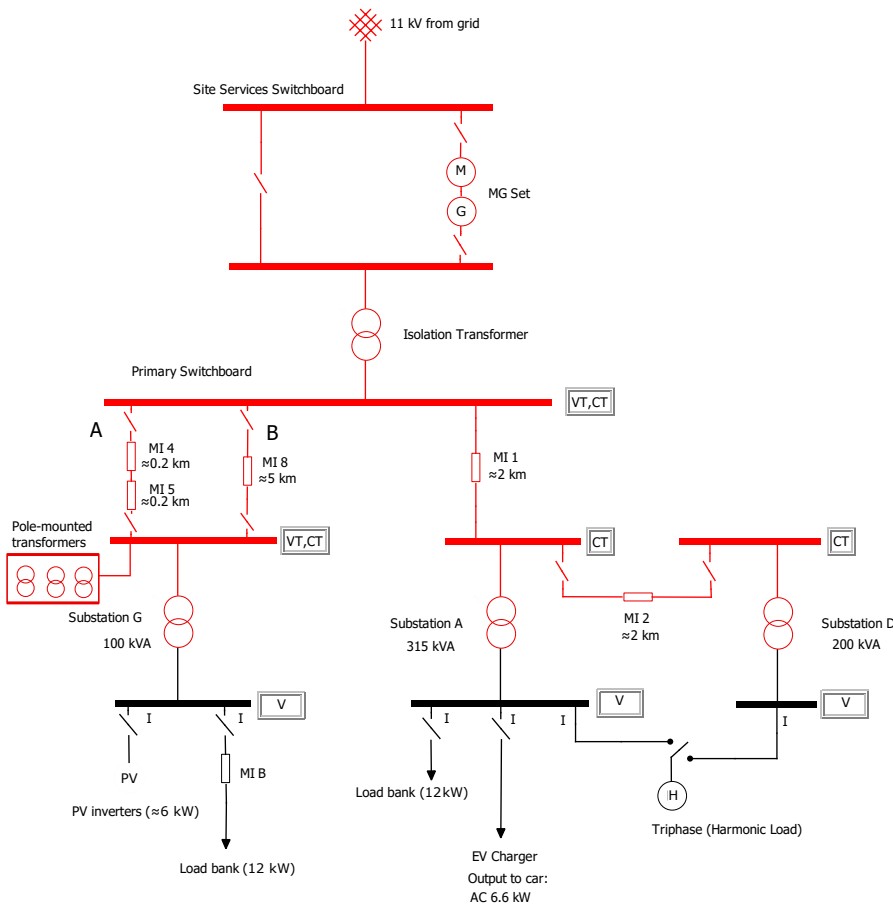

**Figure 1.** One-line diagram of test network.

## 3.2. Measurement System

The acquisition system is built on Beckhoff voltage and current sampling cards and is configured to 5 kHz sampling frequency. Voltage acquisition cards feature an oversampling principle with

16-bit resolution and measurement error not higher than 0.5% relative to full scale value specified for frequencies up to 5 kHz. Current analogue acquisition cards are characterized by differential input and 16-bit resolution with measurement error less than 0.3% relative to full scale. This accuracy is likewise applicable for the 5 kHz range. The monitoring and acquisition of the data are done through TwinCAT software.

On the MV side, there are two voltage measurement points wired to VTs of Substation G and Primary Switchboard (Isolation Transformer). MV currents are measured through every substation as well as through Isolation Transformer. The accuracy class of these CTs is 0.2S providing superior performance at fundamental frequency. Since MV current transducers are typically not characterized for harmonic frequencies, a higher phase shift can be expected at the output. In this paper, we do not use harmonic phase angles of MV side currents. On the LV side, three voltage channels for every busbar are connected directly to the three phases at 250 V. Individual currents of every load are measured by means of Fluke i1000s AC Current Probes providing four-channel current input to sampling cards. The range selector of these current clamps was switched to 10 mV/A sensitivity with minimum measured current of 100 mA and maximum of 100 A. The basic accuracy at this range is 2% of reading for the frequencies up to 5 kHz with a phase shift of no more than 10 degrees. Within the reported frequency range, the linearity of measurements is ensured. On site, all the utilized measurement equipment had valid calibration certificates.

As it can be seen, this measurement system is characterized by modular architecture and capable of recording synchronously voltage and current waveforms. The distributed clock principle with local clock on slave controllers and a master clock ensure the synchronization of all components of the system via unique EtherCAT technology. Additionally, this system is synchronized to the higher-level absolute time making use of external reference based on IEEE1588 PTP. This technique allows all the distributed clocks to be adjusted to a constant offset. By combining these two distinct synchronization routines, it was possible to achieve the distributed clock precision of better than 1 µs.

*3.3. Harmonic Phasor Estimation*

The measurement system provides multi-point, synchronously sampled signals. Due to the presence of time-varying harmonics in the grid, the measured signals are non-stationary in nature. To analyze such non-stationary signals, discrete-time Short-Time Fourier Transform (STFT) is used to perform a joint time-frequency analysis. For discrete-time STFT, the signals $x[n]$ are multiplied by a window $w[n]$ (rectangular/Hann/Hamming, etc.) and Fourier transform (FT) is computed for the resulting signal

$$x_w[n] = x[n]w[n]. \tag{1}$$

If $m$ is the center of the window, then time-frequency representation of the measured signal using STFT is written as [16]:

$$X_{STFT}(e^{j\omega}, m) = \sum_{n=-\infty}^{\infty} x[n]w[n-m]e^{-j\omega n}. \tag{2}$$

The window is then shifted by a fixed amount in time depending on the desired overlap length. The Discrete Fourier Transform (DFT) of the windowed signal $x_w[n]$ is obtained by the convolution of their FTs (windowing theorem) [16]. For a multi-tone windowed signal consisting several harmonics ($\omega_h$) of the fundamental frequency ($\omega_0$)

$$x_w[n] = \left( \sum_h A_h cos(\omega_h \frac{n}{F_s}) \right) w[n], \tag{3}$$

where $A_h$ and $\phi_h$ are the respective magnitude and phase of the constituent harmonics at frequencies $(\omega_h = h\omega_0)$, its DFT is given as:

$$X_{\text{w}}(e^{j\omega}) = X(e^{j\omega}) * W(e^{j\omega})$$
$$= \sum_h \left( X(e^{j\omega_h})W(e^{j(\omega+\omega_h)}) + X(e^{j\omega_h})W(e^{j(\omega-\omega_h)}) \right). \tag{4}$$

where $X(e^{j\omega})$ and $W(e^{j\omega})$ are the FTs of $x[n]$ and $w[n]$, respectively. Thus, the FT of the windowed signal consists of the FT of the window replicated at frequencies $\pm\omega_h$ [17]. Since DFT is obtained at discrete frequencies, the result for frequency $\omega_k$ can be written as:

$$X_{\text{w}}(e^{j\omega})|_{\omega_k} = \sum_h \left( X(e^{j\omega_h})W(e^{j(\omega+\omega_h)}) + X(e^{j\omega_h})W(e^{j(\omega-\omega_h)}) \right)|_{\omega=\omega_k}$$
$$X_{\text{w}}(e^{j\omega_k}) = \sum_h \left( \frac{A_h}{2}e^{j\phi_h}W(e^{j(\omega_k+\omega_h)}) + \frac{A_h}{2}e^{j\phi_h}W(e^{j(\omega_k-\omega_h)}) \right), \tag{5}$$

where the frequency of the bins for $N$ samples are

$$\omega_k = 2\pi k/N \quad \text{for } k = 0, 1, 2, ..., N-1. \tag{6}$$

The positive half of the spectrum can be written as:

$$X_{\text{w}}^+(e^{j\omega_k}) = \sum_h \left( \frac{A_h}{2}e^{j\phi_h}W(e^{j(\omega_k-\omega_h)}) \right) \tag{7}$$

Equation (7) can be interpreted as a filter tuned for favorable response at desired harmonic frequencies $(\omega_h)$. Thus, the harmonic phasors could be calculated using Equation (7) for $\omega_k = \omega_h$. For this measurement campaign, the focused frequencies were the fundamental (50 Hz) and the odd harmonics up to thirteenth harmonic. However, the fundamental grid frequency and thus the harmonic frequencies could vary in real time making $\omega_h$ unknown while the sampling frequency of the measurement set-up remains constant. This lack of coherency between sampling frequency and frequencies $\omega_h$ in the grid measurement data affects the performance of the filters leading to spectral leakage [16]. This causes errors in the magnitude and phase of the estimated harmonic phasors. However, if the unknown grid frequency is estimated, then this frequency and the harmonics can be utilized in analytical expression of the window function to estimate the correct magnitude and phase information of the harmonic phasors. Hence, to minimize the effect of spectral leakage, frequency domain interpolation presented in [16] was used to estimate the correct power-grid frequency and then calculate the corrected magnitude and phase of the harmonic phasors.

For a given harmonic number $h$, in case of deviation of the frequency by $\Delta\omega_h$ such that $\Delta\omega_h = \delta f_r$, actual frequency $(\omega_h)$ from the DFT frequency bins can be given by:

$$\omega_h = f_r(l_0 + \delta). \tag{8}$$

where $f_r$ is the frequency resolution given by $\frac{2\pi}{N}$, $l_0$ is the index of the frequency bin with highest magnitude, $\omega_k = l_0 f_r$, and $|\delta| \leq 0.5$. The deviation in the frequency is determined using the ratio between the two highest DFT components given by:

$$\alpha = \frac{|X_{\text{w}}^+[l_0 + \epsilon]|}{|X_{\text{w}}^+[l_0]|} \tag{9}$$

where $\epsilon$ could be either 1 or $-1$ depending on the position of the second highest DFT bin compared with respect to $l_0$. It can be shown that, for $k = l_0 + \epsilon$,

$$\omega_k - \omega_h = (1 - \delta)f_r. \tag{10}$$

Similarly, for $k = l_0$,

$$\omega_k - \omega_h = -\delta f_r. \tag{11}$$

Utilizing Equations (7), (10), and (11), Equation (9) can be written as:

$$\alpha = \frac{|W(e^{j(\epsilon - \delta)f_r})|}{|W(e^{-j\delta f_r})|}. \tag{12}$$

where the frequency response of the window function is dependent on the type of window used.

The frequency response of a rectangular window of length $N$ samples is given by:

$$W_{\text{rec}}(e^{j\omega}) = e^{-j\omega(N-1)/2}\frac{\sin(\omega N/2)}{\sin(\omega/2)}. \tag{13}$$

A $\delta$-$\alpha$ look-up table was created using Equation (12) for uniformly spread out values of $\delta$. A Hann window was used in the process whose frequency frequency response is given by:

$$W_{\text{hann}}(e^{j\omega}) = 0.5W_{\text{rec}}(e^{j\omega}) - 0.25W_{\text{rec}}(e^{j(\omega - \omega_1)}) - 0.25W_{\text{rec}}(e^{j(\omega + \omega_1)}), \tag{14}$$

where $\omega$ is $(\epsilon - \delta)f_r$ and $\omega_1$ is $f_r$.

In the look-up table, the values of calculated $\alpha$ was paired with the closest value of $\delta$ to determine the deviation in the frequency. The actual frequency ($\omega_h$) is calculated as:

$$\omega_h = \omega_k \pm \delta f_r. \tag{15}$$

From Equation (7), the ratio of magnitudes for one half of the spectrum can be written as:

$$\frac{|X_{\omega_h}|}{|X_{\omega_k}|} = \frac{|W(e^{j\omega_h})|}{|W(e^{j\omega_k})|}. \tag{16}$$

Using the relationship between $\omega_h$ and $\omega_k$ from Equation (15), the corrected phasor magnitude is calculated using Equation (16) as:

$$|X_{\omega_h}| = |X_{\omega_k}|\frac{|W(e^0)|}{|W(e^{j\delta f_r})|}. \tag{17}$$

Similarly, the phase at two frequencies are related as:

$$\arg\{X_{\omega_h}\} = \arg\{X_{\omega_k}\} \pm \arg\{W(e^{j\delta f_r})\}, \tag{18}$$

where

$$\arg\{W(e^{j\delta f_r})\} = \arg\{e^{-j\delta\pi(N-1)/N}\}, \\ = \delta\pi(N-1)/N. \tag{19}$$

The presented interpolated-DFT method was used to estimate accurate harmonic phasors in real grid conditions. The following subsection presents the different test cases used in the study.

*3.4. Test Cases*

The harmonic propagation study includes two distinct packages. In Package A, an MG set is used to energize the network with ideal sinusoidal voltage supply. Every harmonic load is connected separately at its designated position. The individual current waveforms and synchronized voltage waveforms are measured at various points of the network. This test concludes with putting all the loads into the operation simultaneously and recording all the electrical parameters synchronously.

In test package B, the network is energized by the public grid and at the first stages the same scenarios as in package A are tested. Furthermore, an equal share of resistive load banks is connected to the substations G and A. At the next stage, the topology of the network was changed at the MV level by bypassing the impedance section A through section B. Finally, the last change in the configuration involved replacement of the Triphase-emulated harmonic load from Substation A to Substation D. The test mapping with change variables is shown in Table 1.

**Table 1.** Experimental scenarios.

| Test Sequence | Power Supply | PV Setpoint | EV Charger | Triphase | Load Banks | Topology |
|:---:|:---:|:---:|:---:|:---:|:---:|:---:|
| A.1 | MG | 7.5 A | Disconnected | Disconnected | Disconnected | A |
| A.2 | MG | Disconnected | 32 A | Disconnected | Disconnected | A |
| A.3 | MG | Disconnected | Disconnected | Substation A | Disconnected | A |
| A.4 | MG | 7.5 A | 32 A | Substation A | Disconnected | A |
| B.1 | Grid | 7.5 A | Disconnected | Disconnected | Disconnected | A |
| B.2 | Grid | Disconnected | 32 A | Disconnected | Disconnected | A |
| B.3 | Grid | Disconnected | Disconnected | Substation A | Disconnected | A |
| B.4 | Grid | 7.5 A | 32 A | Substation A | Disconnected | A |
| B.5 | Grid | 7.5 A | 32 A | Substation A | 24 kW | A |
| B.6 | Grid | 7.5 A | 32 A | Substation A | 24 kW | B |
| B.7 | Grid | 7.5 A | 32 A | Substation D | 24 kW | B |

Topology A—mock impedances MI 4 and MI 5; topology B—mock impedance MI 8.

## 4. Results

The amount of measurement data as the outcome of this experiment is significant, giving measurement windows of 10 min for every test sequence within package B and in an average of 2 min to 3 min for testing scenarios of package A. Typically, around 30 synchronized channels of raw waveforms were recorded for each test. For representing results of this work, we chose to process only one phase (A) for every measurement in question. The signal processing method of choice is FFT with 10 cycles rectangular window. For package A, only one selected window was used for the graphical representation of the results. As it was stated in the Section 3.1 when energizing the network with MG-set, a non-distorted voltage at the terminals of Isolation Transformer is ensured. This allows for focusing on the harmonic processes originating from the downstream network on the customer side since background voltage harmonics cannot penetrate from MV upstream network via MG-set. It was therefore concluded that one 10-cycle window can be sufficient for analyzing harmonic distortion when no fluctuating background voltage harmonics are present in the test grid.

On the contrary, with grid power supply, it was important to demonstrate time evolution of the harmonics. For this purpose, every 10 min waveform was fully processed and 10-cycle RMS magnitudes were collected into coherent plots with 600 s duration time. This technique allowed for evaluating harmonic trends being a function of fluctuating voltage background distortion. For one specific test case B.4, we introduced additionally a numerical analysis based on the estimated synchronized harmonic phasors. Only results from the selective cases will be demonstrated here.

*4.1. Separate Operation of the EV Charger*

The analysis begins by examining EV charger current emission and by assessing respective voltage distortion under sinusoidal power supply conditions. Figure 2 shows that current spectral components

of the EV charger for this case are characterized by high values of harmonics at 1150 Hz and 1250 Hz, which are 0.71 A and 1.24 A, respectively. Moreover, 150 Hz and 250 Hz spectral components exhibit 0.35 A while a 350 Hz harmonic component counts some 0.42 A. Finally, lesser levels can be observed for frequencies between 450 Hz and 1050 Hz.

By analyzing synchronized voltage spectral components presented in Figure 2, one can notice a coherent picture, where at 1150 Hz and at 1250 Hz harmonic magnitudes are 1.90 V and 2.80 V, respectively. For the rest, only harmonics at 250 Hz, 350 Hz, and 650 Hz are clearly visible with RMS magnitudes of 0.67 V, 1.22 V, and 0.65 V, respectively.

On the other hand, by energizing test networks by means of a public grid, a clear change of the harmonic profiles can be observed in Figure 3. An increase of 350 Hz and 550 Hz current components up to 0.51 A and 0.246 A, respectively, coincided with the rise of voltage at 350 Hz up to 1.64 V and substantial decrease of 650 Hz voltage harmonic which counted 0.284 V.

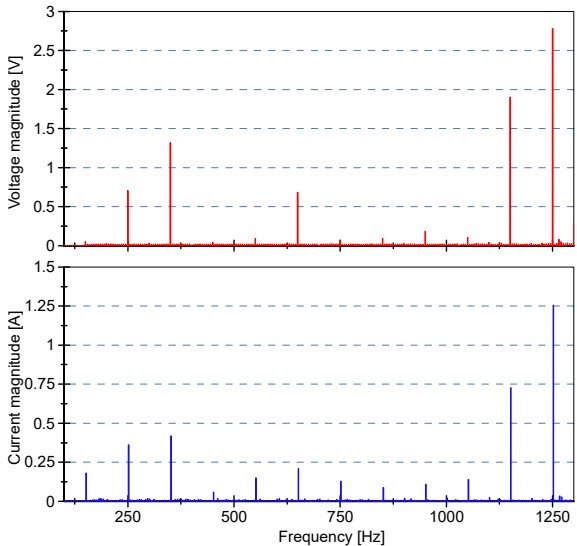

**Figure 2.** EV charger harmonic currents and voltages on the LV side of Substation A. Test sequence A.2—MG set supply.

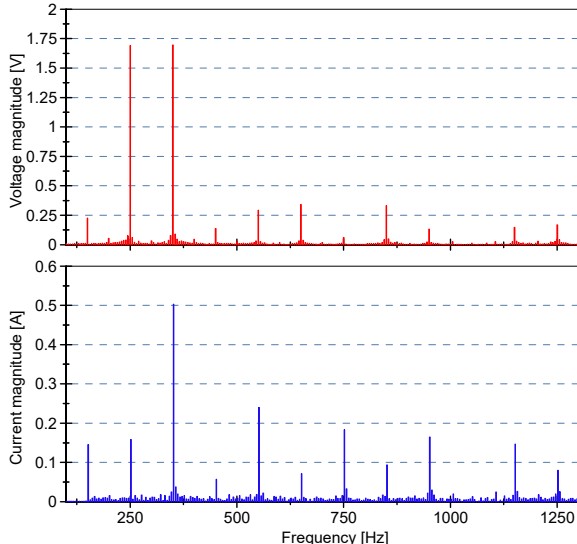

**Figure 3.** EV charger harmonic currents and voltages at LV side of Substation A. Test sequence B.2—public grid supply.

A value drop at 1150 Hz and 1250 Hz is quite remarkable with voltage and current levels at these frequencies being negligible. One of the underlying reasons explaining this sudden reduction of the aforementioned voltage frequency components is the different values of short-circuit power between cases A.2 and B.2. The power supply via public grid was characterized by 23.4 MVA short-circuit power whilst MG set's short-circuit power was equal to 8.3 MVA. On the contrary, a major difference can be noticed for 250 Hz with voltage magnitude as high as 1.68 V but in comparison with Figure 2 significantly smaller current of 0.14 A.

Furthermore, for test scenario A.2 (MG set supply), a set of synchronous voltage measurements at MV and LV sides of Substation G was processed and results are shown in Figure 4. At MV busbar, we recorded 15 V and 32 V at 250 Hz and at 350 Hz, respectively, and 16 V at 650 Hz. Moreover, the largest voltage harmonic distortion values were observed at 1150 Hz and 1250 Hz with 37 V and 63 V, respectively. Looking at the voltage harmonic values at LV test bay of Substation G, it is interesting to note differences in values at frequencies 1150 Hz and 1250 Hz when comparing to these from Figure 2. The levels of voltage distortion at these frequencies are lesser and equal to 1.60 V and 2.26 V for 1150 Hz and 1250 Hz, respectively.

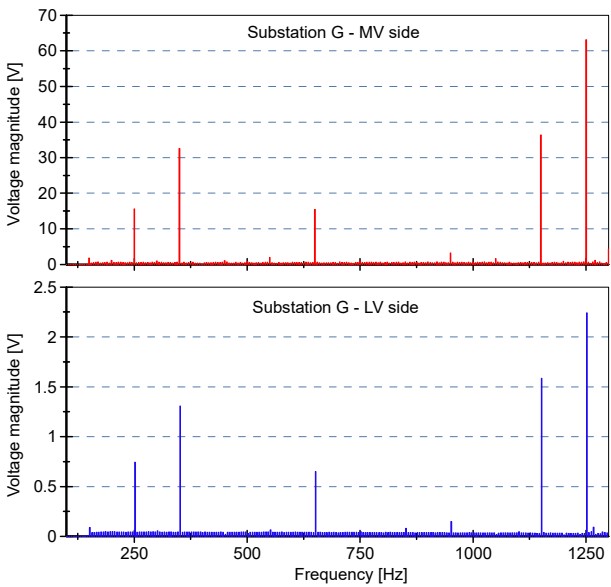

**Figure 4.** Harmonic voltages at MV and LV sides of Substation G. Test sequence A.2 — MG set supply.

As it can be deduced from the presented results, there exist visible differences in harmonic current emissions of an EV charger when exposed to different levels of background voltage distortion (cases A.2 and B.2). To be able to interpret results, it was decided to measure harmonic current injections of this device by connecting it directly to a programmable power source. By supplying ideal sinusoidal voltage waveform to the input terminals of the EV charger, its load current waveform was recorded and processed with an FFT algorithm. This procedure is similar to the one usually performed during factory testing of power electronic interfaced equipment when the final THD value of the device under testing is then included in technical specifications. Figure 5 shows the FFT output of EV charger harmonic current emissions. When comparing to values from Figure 2, it can be concluded that the major differences are to be observed at 650 Hz, 1150 Hz, and 1250 Hz. At these frequencies, the natural emissions of this device are negligible, which brings the conclusion about harmonic interaction between already existing in the test network voltage distortion and harmonic current injections of EV charger. The measurement process with regard to this phenomena and some implications are best described in [11,18,19]. Moreover, the same phenomena are observed for frequency components at 250 Hz and 350 Hz—however, with current injections at these frequencies slightly higher when connected to the test grid.

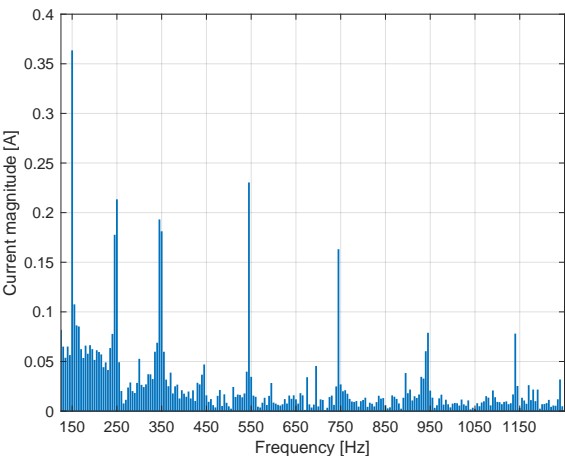

**Figure 5.** Harmonic emission of EV charger under ideal sinusoidal supply conditions.

Thus, a direct comparison of Figures 2, 4, and 5 allows for inferring that voltage distortion recorded at different points could not be caused by the EV charger itself but rather propagated from the path magnetizing circuit-MV side Substation G-MV side Substation A. This conclusion can be supported by measuring synchronized spectra of magnetizing current produced by pole-mounted transformers versus spectra of the MV current through Substation A. This graph is shown in Figure 6. Based on this graph, it can be inferred that harmonic components of magnetizing current had a dominant influence on the voltage distortion in the system. Therefore, this current impacted MV voltage at Substation G and propagated down to its LV side, where reflected MV distortion patterns are clearly visible. Additionally, the synchronized data allow for drawing interesting conclusions with respect to Substation A. At frequencies 1150 Hz and 1250 Hz, we can notice that voltages at LV side are substantially higher in comparison with ones at Substation G. This is not only related to a larger system harmonic impedance with underground cable MI 1 being 5 times longer than MV cable connecting Substation G (in total 400 m), but also to a fact that, at these frequencies, an EV charger's emissions were relatively high, which resulted in larger harmonic voltage drop over impedance at these frequencies (as Figure 6 supports).

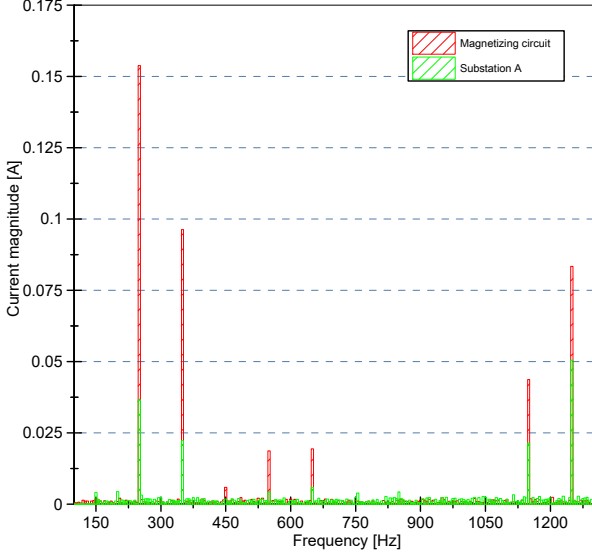

**Figure 6.** Comparison of magnetizing current spectra versus MV current through Substation A. Test sequence A.2 — MG set supply.

Furthermore, after analyzing the results presented in Figure 3, a conclusion can be drawn about the influence of harmonic voltage propagated from the upstream grid. A notable observation is a dissolution of voltage harmonics at 1150 Hz and 1250 Hz which again provoked change of harmonic currents at these frequencies. On the other hand, an increase of voltages at 250 Hz and at 350 Hz coincided with the change of EV charger current at these frequencies. While, as it was mentioned before, higher frequency order harmonic voltages reduced because of the higher short-circuit power capacity of the public grid, the change of spectral components at 250 Hz and 350 Hz was provoked by propagated background distortion interacting with the operation of the studied harmonic sources.

### 4.2. Multiple Harmonic Sources under Sinusoidal Power Supply

The simultaneous operation of devices injecting harmonics into grid is analyzed in this section. The loads studied in this test case are PV inverter, EV charger, and synthetic harmonic load emulated by Triphase. The system is energized by MG power supply set (sequence A.4). In Figure 7, synchronized voltage spectral components extracted from all points available for voltage measurements are shown. This plot contains components which were otherwise not present in previous cases. The drastic increase of voltages up to 30 V at 850 Hz and up to 22 V 950 Hz at MV side is clearly visible. Additionally, at 550 Hz, the recorded harmonic voltage value was approximately 11 V. It is obvious to note that no substantial difference was observed between voltages at Primary Switchboard (Isolation Transformer) and MV side of Substation G. The reason for this is relatively small electrical distance between these two points.

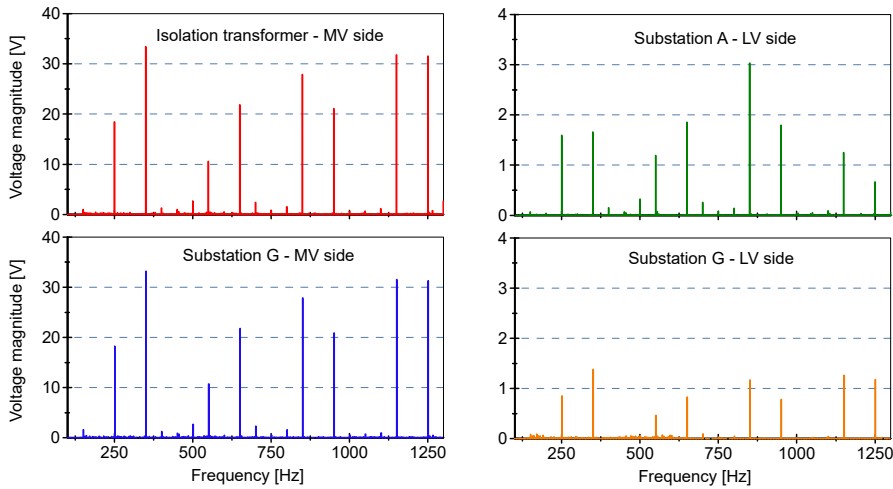

**Figure 7.** Harmonic voltages at multiple points. Test sequence A.4—MG set supply.

In order to gain insight into the reasons behind such harmonic response of the studied network, the synchronized substation currents were processed and results are demonstrated in Figure 8. The currents through Substation A are significantly higher than through Substation G for every frequency except 1150 Hz and 1250 Hz harmonic components. By looking at individual currents of every connected harmonics source, more details can be revealed. In Figure 9, the synchronized current spectra of PV inverter and EV charger are presented. Whilst the emissions of PV are blurred with significant amount of subharmonics, the harmonic currents of EV charger coincide with measured voltages.

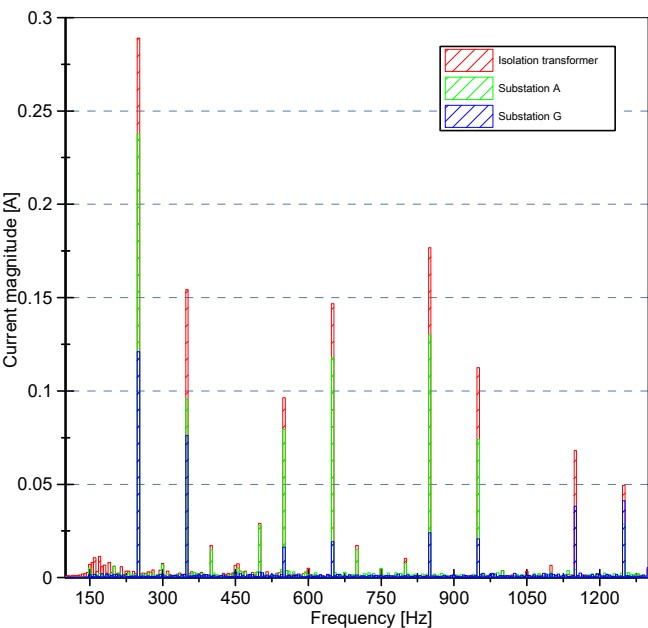

**Figure 8.** MV harmonic currents through transformers. Test sequence A.4—MG set supply.

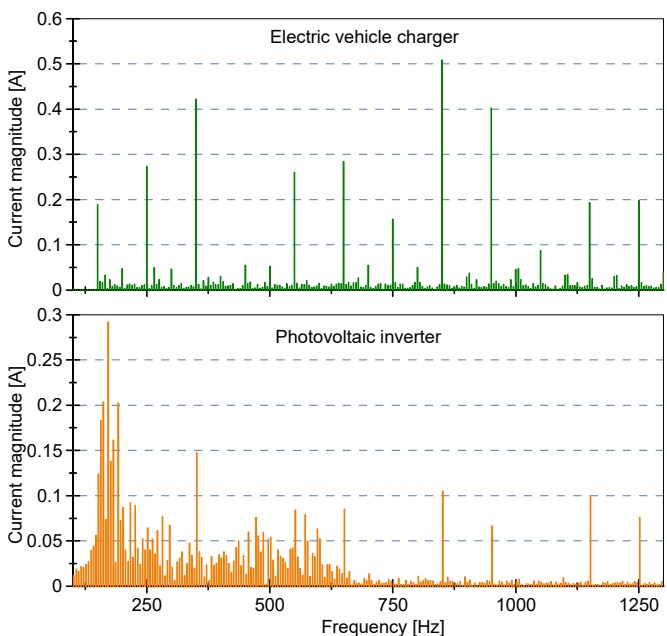

**Figure 9.** Individual harmonic currents of PV and EV chargers. Test sequence A.4—MG set supply.

Furthermore, the Triphase connected to substation A was programmed as a substantially large source of harmonic current injections with a purpose of acting as reference device with fixed harmonic emissions. The values of harmonic currents for this load are presented in Table 2 for frequences 150 Hz to 1050 Hz. It is worth noting that harmonic phase angles for every order were set to 0.

**Table 2.** Currents of emulated harmonic source (in A).

| 3rd | 5th | 7th | 9th | 11th | 13th | 15th | 17th | 19th | 21st |
|-----|-----|-----|-----|------|------|------|------|------|------|
| 1.055 | 6.07 | 2.312 | 0.225 | 2.167 | 2.805 | 0.588 | 3.324 | 1.644 | 0.091 |

Next, the measured harmonic currents of this device are shown in Figure 10. In comparison with Figure 7, the conclusions can be drawn that this emulated load acted as dominant source of harmonics provoking large harmonic voltage drops at 550 Hz, 850 Hz, and 950 Hz on the LV side of Substation A. This distortion in turn propagated towards the MV system and down to the LV side of Substation G. Moreover, while the 5th harmonic current injected by Triphase is substantial (6.07 A), it did not cause a significant voltage drop at this frequency at a medium voltage level. The reason for this is relatively small low-voltage impedance at this frequency which resulted on average in some 0.5 V of harmonic voltage drop at an LV level. Propagating further upstream, the resulting harmonic current phasor interacting with impedance angle at this particular frequency produced additionally about 3 V.

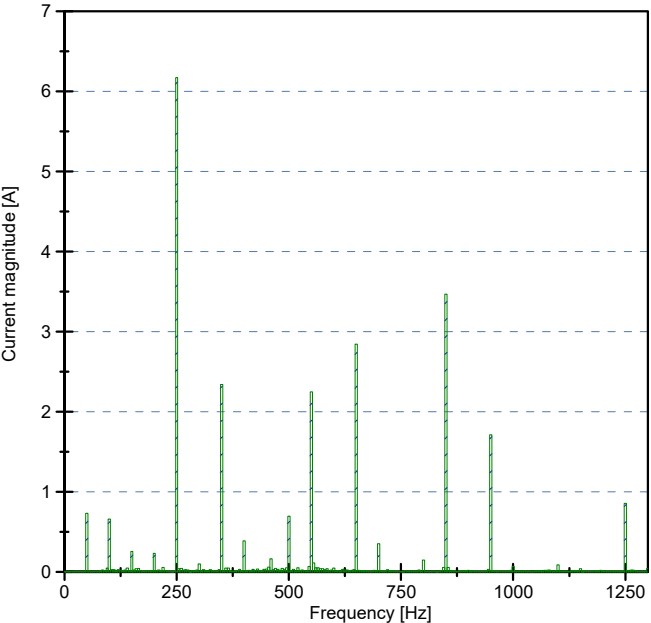

**Figure 10.** Currents of emulated harmonic source.

It is interesting to observe that this complex interaction between harmonic current phasors and system impedances provoked a significant decrease of voltage at 1250 Hz from 63 V (case A.2) to about 32 V. Since the emulated harmonic load was not programmed to inject current at this frequency, we shall look again at Figures 8 and 9 in order to evaluate harmonic current emissions of PV and EV chargers. While currents through Substation G are slightly higher, Figure 9 suggests that both PV inverters and EV chargers can be deemed to be responsible for the fraction of emissions at 1250 Hz. A notable observation is related to 1250 Hz current component of Triphase visible in Figure 10. Since no injection was set at this frequency, we can conclude that this component resulted from the absorption of harmonics by the filter at the output stage of this device.

Finally, in Figure 7, we observe that propagated voltage distortion at LV side of Substation G settles to lower values when compared to the LV side of Substation A. As it was discussed before, the fundamental reason for this is lesser system impedance governed by short underground cable connection. An exception here is voltage harmonic at 1250 Hz recorded on the LV side of Substation G.

*4.3. Multiple Harmonic Sources Energized by Public Grid*

The operation of the public grid is characterized by the presence of time-varying background harmonic voltages. This alters the harmonic behavior of the loads studied in this work and influences the final values of voltage distortion. To illustrate this case, the outcome of FFT analysis for selective harmonics collected in the form of synchronized time series is presented in Figure 11. A highly

fluctuating behavior of voltages at 250 Hz and 350 Hz can be observed with some periodic patterns visible at frequencies 550 Hz and 650 Hz.

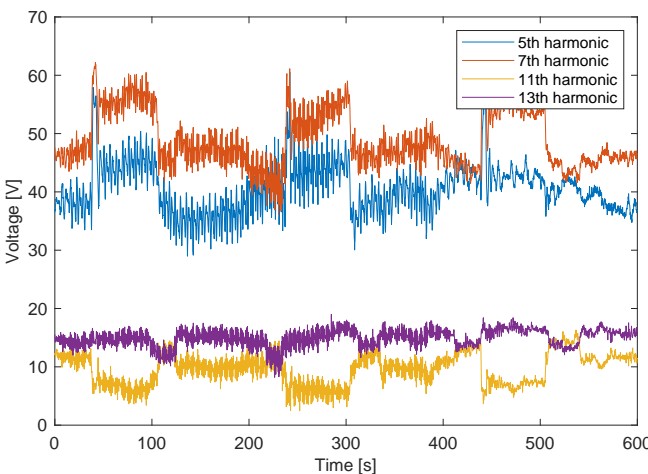

**Figure 11.** Isolation transformer harmonic voltages. Test sequence B.4—public grid supply.

Furthermore, in Figures 12, and 13, synchronized time series of harmonic currents through distribution transformers are shown. It can be noted that these currents exhibit also time-varying behavior with levels of some of the harmonics changed in comparison with case A.4.

Looking at the harmonic voltages at low-voltage test bays (Figures 14 and 15), we observe a remarkable difference between these two points. With harmonic voltage at 250 Hz at Substation A as high as 3.7 V against a maximum of 2.7 V at Substation G and significantly higher values of voltages at 550 Hz and 650 Hz.

Moreover, the final value of voltage distortion at MV level is the complex sum of background distortion transferred from upstream network and voltage drop produced by combined harmonic current phasor flowing through harmonic system impedance. A direct comparison between cases represented by Figures 8, 12 and 13 can lead to inconsistent conclusions about the nature of harmonic processes occurring in the grid; therefore, a quantitative method to evaluate the undergoing phenomena is required.

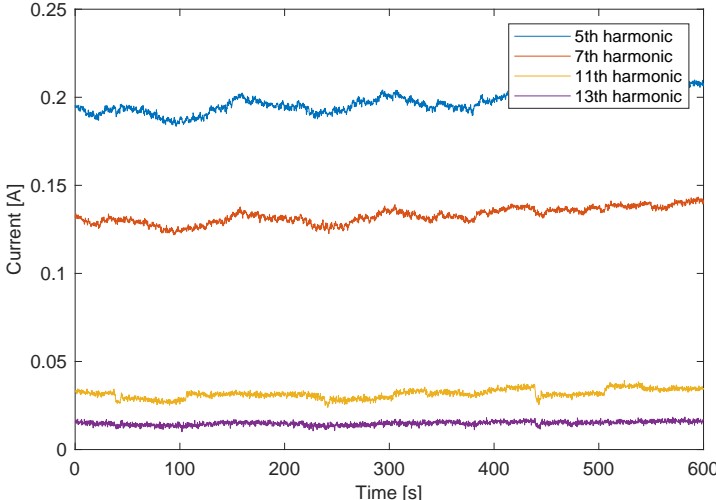

**Figure 12.** Harmonic currents through Substation G. Test sequence B.4—public grid supply.

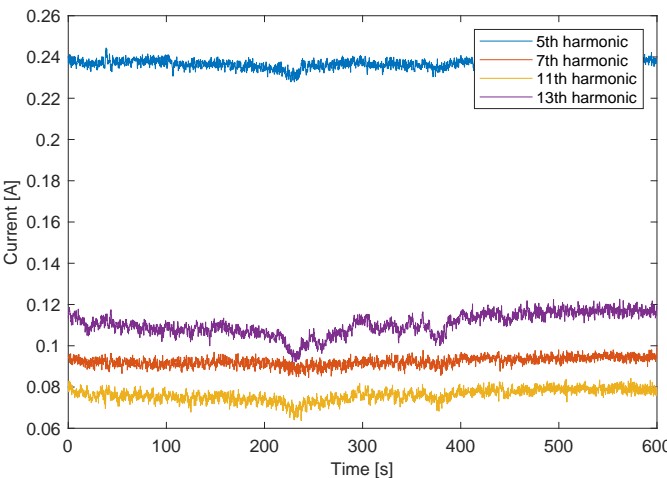

**Figure 13.** Harmonic currents through Substation A. Test sequence B.4—public grid supply.

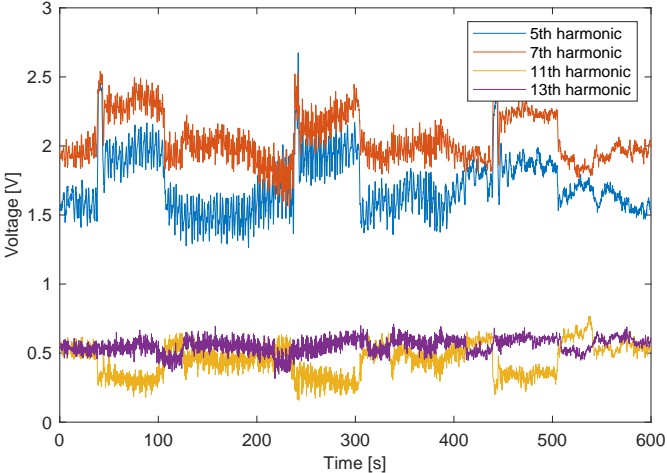

**Figure 14.** Harmonic voltages at LV side of Substation G. Test sequence B.4—public grid supply.

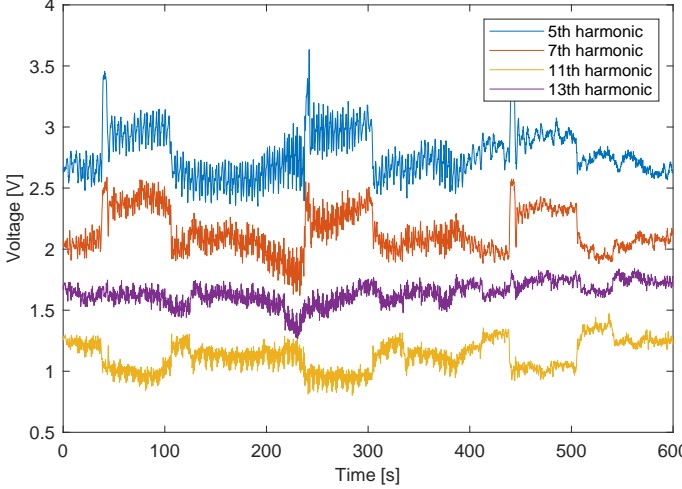

**Figure 15.** Harmonic voltages at LV side of Substation A. Test sequence B.4—public grid supply.

Synchronized harmonic phasors present a powerful tool for analyzing harmonic data. In this paper, an example is given on how to estimate LV harmonic contributions based on the Voltage Harmonic Vector Method [20]. For the sake of clarity, only harmonic components at 250 Hz and 350 Hz were considered in the calculations.

The calculations according to the Voltage Harmonic Vector Method require employment of utility harmonic impedance. In this work, we assume the MV/LV transformers as components dominating the system LV harmonic impedance. Based on the datasheets, the fundamental frequency reactance was estimated to 0.05 $\Omega$ and 0.0179 $\Omega$ for Substation G and Substation A, respectively. The harmonic impedances at 250 Hz and 350 Hz were then recalculated based on simple theoretical derivations providing proportional linear approximations for the frequency-dependent inductance. According to the Voltage Harmonic Vector Method, the following applies:

$$\underline{U}_{U-h} = \underline{U}_{\text{PCC}-h} - \underline{I}_{\text{PCC}-h} \cdot \underline{Z}_{U-h}, \tag{20}$$

where $\underline{U}_{U-h}$ is utility voltage representing background harmonic voltage source, $\underline{U}_{\text{PCC}-h}$ is harmonic voltage measured at LV side of respective substation, $\underline{I}_{\text{PCC}-h}$ vectorial sum of individual injection currents, and $\underline{Z}_{U-h}$ is utility harmonic impedance. The harmonic impedance of transformer can be calculated as:

$$\underline{Z}_{U-h} = h \cdot X_f \tag{21}$$

where $h$ is a harmonic order and $X_f$ is a transformer reactance at fundamental frequency.

Therefore, based on the Equation (22), the following holds:

$$\underline{U}_{\text{emission}} = \underline{U}_{\text{PCC}-h} - \underline{U}_{U-h} \tag{22}$$

Figures 16 and 17 demonstrate Substation A estimated voltage harmonic phasors for background distortion and voltage emission, respectively. The individual phasors of harmonic currents of equipment connected to the Substation A are shown in Figures 18 and 19.

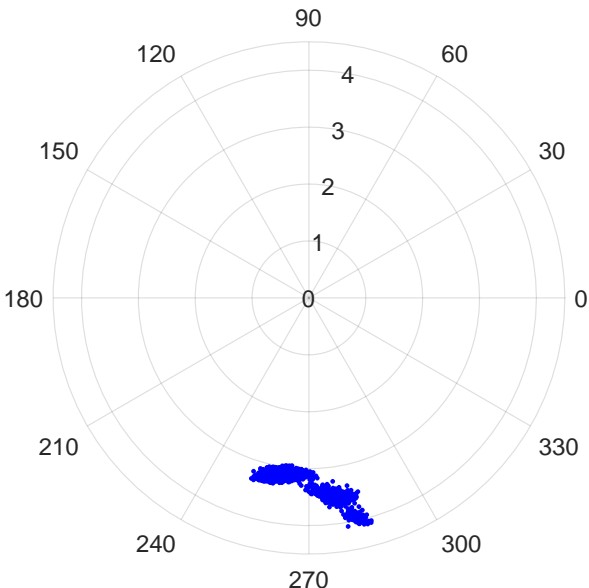

**Figure 16.** 5th harmonic background voltage phasors at LV side of Substation A (in volts). Test sequence B.4—public grid supply.

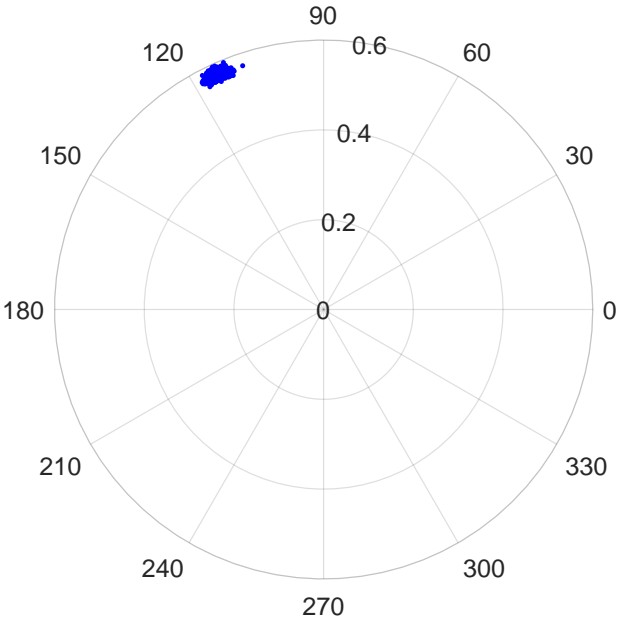

**Figure 17.** 5th harmonic voltage emission phasors at LV side of Substation A (in volts). Test sequence B.4—public grid supply.

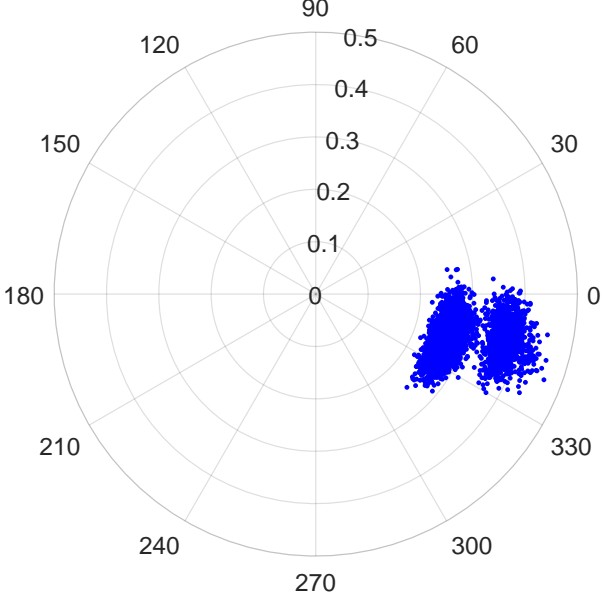

**Figure 18.** 5th harmonic current phasors of EV at Substation A (in amperes). Test sequence B.4—public grid supply.

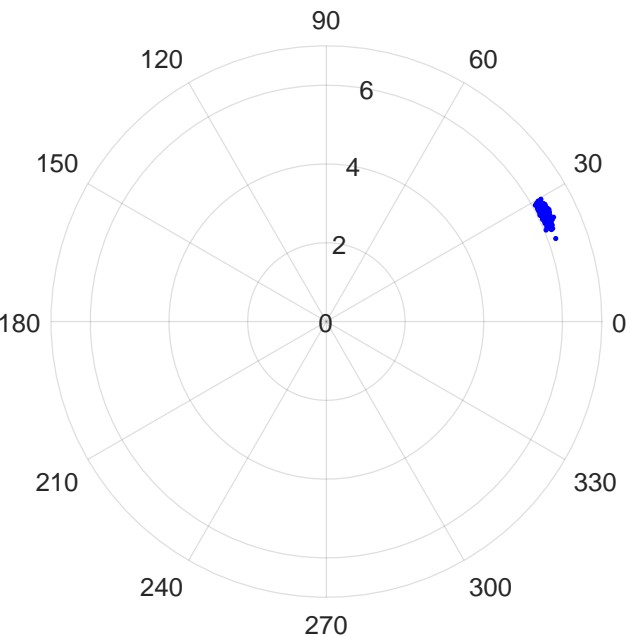

**Figure 19.** 5th harmonic current phasors of Triphase at Substation A (in amperes). Test sequence B.4—public grid supply.

Figure 20 shows the result of harmonic current vectorial summation at LV side of substation A. It can be observed that, in that case, emulated harmonic load acts as the dominant source of harmonic current at 250 Hz. The harmonic voltage drop produced by this harmonic current (Figure 17) ensures a certain level of harmonic cancellation when added to the estimated voltage background phasors (Figure 16) with final measured values settled below the initial background levels (Figure 21).

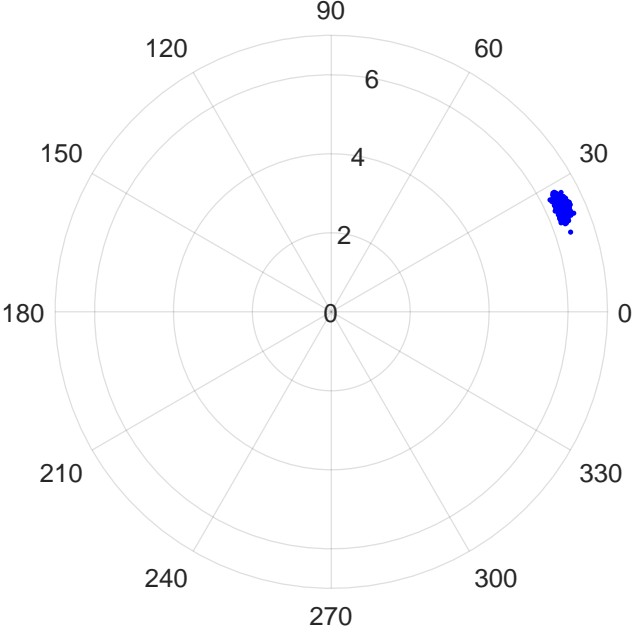

**Figure 20.** 5th harmonic current phasors (vectorial sum) at Substation A (in amperes). Test sequence B.4—public grid supply.

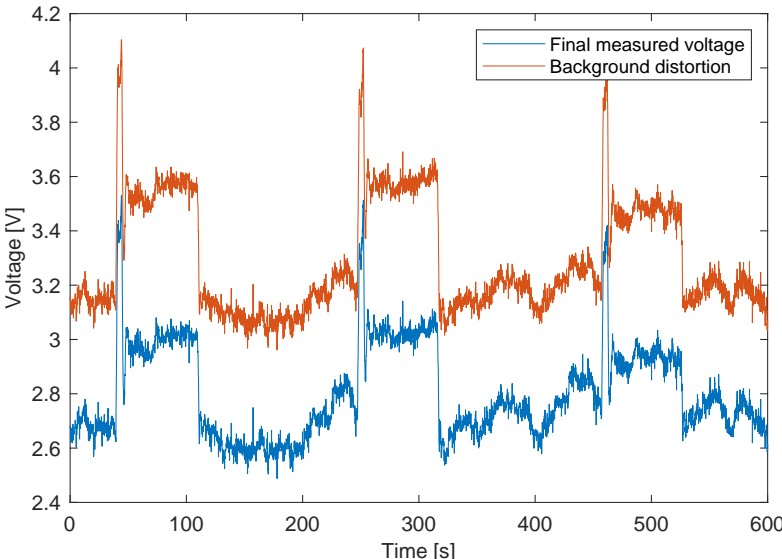

**Figure 21.** 5th harmonic voltages (RMS) at Substation A. Test sequence B.4—public grid supply.

Furthermore, an example of harmonic processes taking place synchronously for 350 Hz harmonic at Substation G is shown in Figures 22 and 23. Despite the dynamic nature of PV power, it is to observe that harmonic voltage emissions phasors at 350 Hz are localized at the 1st quadrant of unity pane. Given the phasors of voltage background distortion located at the 4th quadrant, the conclusions that a small harmonic cancellation took place within this specific substation can be made.

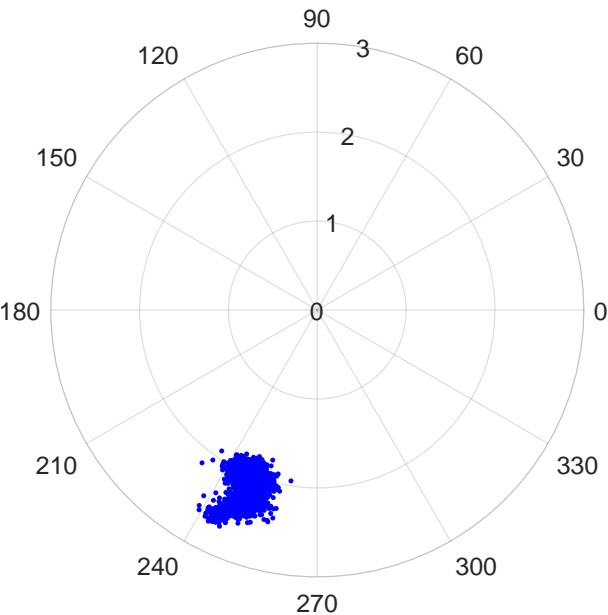

**Figure 22.** 7th harmonic background voltage phasors at LV side of Substation G (in volts). Test sequence B.4—public grid supply.

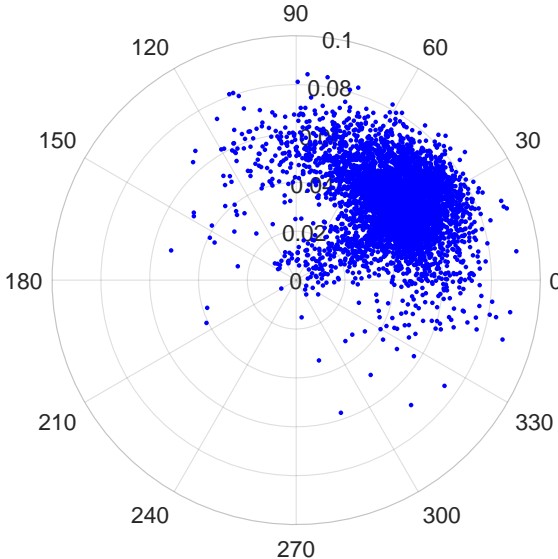

**Figure 23.** 7th harmonic voltage emission phasors at LV side of Substation G (in volts). Test sequence B.4—public grid supply.

Finally, a small selection of average quantitative results derived by means of synchronized phasors and Voltage Harmonic Vector Method is collected in Table 3.

Furthermore, it is interesting to observe the time-evolving harmonic current behavior of EV charger demonstrated in Figure 24. The harmonic injections of this type of power-electronic devices are highly sensitive to the changes of applied harmonic background voltage, both in terms of magnitude and phase angles. This effect is attributed to the underlying topology of the equipment, namely to the type of power factor correction circuit [11,19]. As it can be seen from synchronized harmonic trends, for every studied frequency, the current injections of EV charger exhibit nearly linear behavior in response to the background voltage distortion. This phenomenon renders such multi-point harmonic measurements to be important in an attempt to identify sources of disturbances and to quantify harmonic emissions.

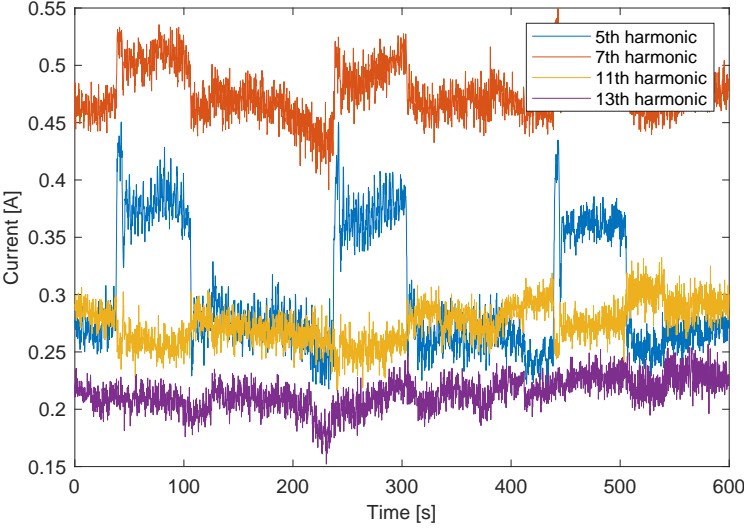

**Figure 24.** EV charger harmonic currents. Test sequence B.4—public grid supply.

**Table 3.** Average RMS magnitude of harmonic voltage emissions per substation.

| Harmonic Frequency, Hz | Emission Voltage Sub A, V | Emission Voltage Sub G, V |
|---|---|---|
| 250 | 0.574 | 0.043 |
| 350 | 0.232 | 0.054 |

*4.4. Impact of Harmonic Impedance on Voltage Distortion Levels*

In conventional power quality analysis, it is a common practice to omit influence of linear loads operating at nearly unity power factor. Such loads, however, by means of resistive nature act as a shunt impedance connected to the LV side of distribution substation. This shunt impedance in turn decreases proportionally the system harmonic impedance providing the low-impedance path for harmonics. Theoretically, this can lead to the decrease of THD levels. This effect is best described in literature as "damping".

The performed experiments, however, demonstrate that in some cases this conclusion can be somewhat too optimistic. In this test sequence (B.5), we connected 24 kW of resistive loads equally divided between both distributed substations and evaluated THD at LV test bays both before and after this operation has been performed. In Figure 25, the results of this experiment are shown in the representative form of box plots rendered over the whole duration of the signals. As it can be seen for both substations, the median levels (red horizontal line) of THD are actually higher after connecting the linear load banks. Here, it is important to note that, while conventional power quality studies and computer simulations (mainly because of the nature of used models of harmonic loads) rely exclusively on calculations based on magnitudes of currents and voltages, the harmonic processes in real grid scenarios are much more complex. Depending on the phase angles of harmonic current and voltage phasor (given that we did not have any control over background distortion) and new value of impedance, this interaction in some cases can lead to the rising levels of final THD values, even when expected otherwise.

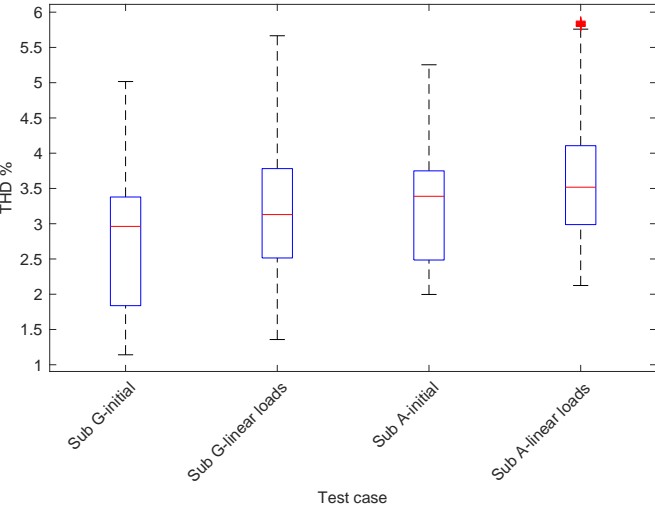

**Figure 25.** Influence of linear load banks on LV-side THD levels. Test sequence B.5—public grid supply.

Next, a dynamically changing topology at MV levels is considered to be a known issue for harmonic analysis. The reconfiguration of the network occurring at any given time can significantly alter the voltage distortion values and if not considered properly in subsequent simulations can lead to the erroneous outcome. As per test scenario B.6, we dropped the short cable connecting Primary Switchboard to Substation G and engaged cable B with equivalent electrical distance of about 5 km (see Figure 1). After recording synchronized voltage waveforms at MV busbars of both Substation G and Primary Switchboard (Isolation Transformer), the THD values were collected in the form of

box plot and these are presented in Figure 26. While the spread of THD values became larger, we can clearly observe the increase of median values and, in this case, increased harmonic impedance at the MV level contributed to larger values of voltage distortion.

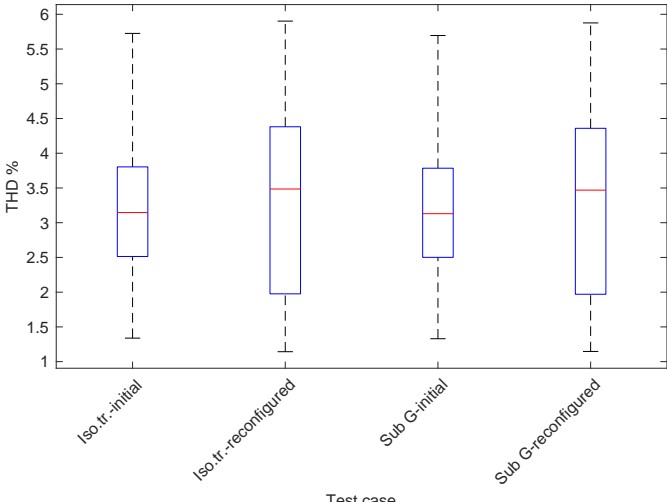

**Figure 26.** Impact of topology change on MV-side THD levels. Test sequence B.6—public grid supply.

*4.5. Load Separation*

The final case studied in this work concerns a harmonic response of the system with an additional distribution transformer. To evaluate the differences, we energized Substation D by means of an underground cable with its sending end connected to the busbar of Substation A (see Figure 1). The estimated length of this cable connection is about 2 km. Next, a reference Triphase harmonic load was moved from Substation A to Substation D. In Figure 27, we only present results for 5th harmonic voltage measured synchronously at all three LV test bays.

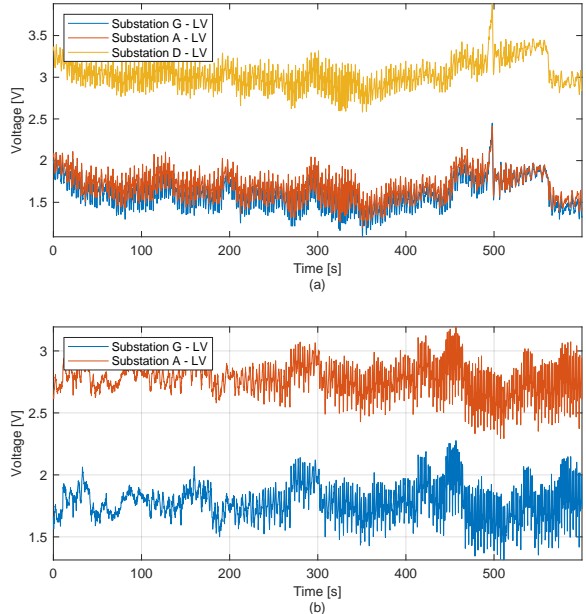

**Figure 27.** Change of 5th harmonic voltages as result of load separation. Test sequence B.7—public grid supply. (**a**) triphase-emulated harmonic load connected to Substation D; (**b**) triphase-emulated harmonic load connected to Substation A.

Whilst harmonic voltage at Substation A dropped after removing dominant injection source, we observe that 250 Hz harmonic trend measured at Substation D is visually higher in comparison with local harmonic voltage values when a disturbance source was connected to the distribution Substation A. Naturally, background harmonic levels also evolved in time, but, certainly, the additional impedance of 2 km cable equivalent boosted harmonic voltages at this substation. Furthermore, the transformer of Substation D is characterized by 0.0295 $\Omega$ fundamental frequency impedance.

## 5. Conclusions

The results of the research project focused on the study of the phenomenon of harmonic distortion propagation through distribution networks were presented. The methodology included processing of synchronized waveforms at both MV and LV levels of the proposed experimental power network. Several MV/LV substations were involved in the experiment. The harmonic impact of PV inverter, EV charger and emulated injection source was the focus of this study. It was shown that utilization of time-synchronized waveform recorders with high accuracy at multiple points of the grid introduces a unique and powerful opportunity to study basic mechanisms governing transfer of harmonic components such as: diversity and harmonic cancellation, harmonic interaction, and attenuation and influence of system and load impedance.

Based on the performed measurements, the global synchronization of current and voltage samples proved to be effective in evaluating diversity phenomena of harmonic currents emitted by various devices. Results show that, depending on the levels of background distortion, current emissions can exhibit cancellations and final voltage distortion value can deviate from anticipated patterns. This fact was also studied and confirmed with provision of harmonic phasors estimated by means of a specific phasor estimation algorithm taking into account fluctuations of fundamental frequency of supplied voltage. Furthermore, taking into consideration summation of harmonics, synchronized voltage and current spectra allowed for deducing conclusions about dominant sources of the distortion and make some judgment about the path of harmonic propagation. As such, one of the important conclusions based on this work is that, with synchronized measurements, it is possible to recreate the impacts on harmonic distortion by calculation, even with network simplifications— so the calculation can be tested and validated in this way as well.

Additionally, based on the executed experiments, system topology changes were considered to be an important factor governing phenomena of harmonic distortion propagation. The effect was also facilitated by the fact that power system components (cables and transformers) exhibit certain frequency-dependence at higher frequencies.

**Author Contributions:** Conceptualization, S.C.; Data curation, A.D.; Formal analysis, S.B. and V.Ć.; Funding acquisition, S.B. and V.Ć.; Methodology, S.B., R.S.S., and S.C.; Project administration, S.C. and A.D.; Resources, A.D.; Software, A.D.; Supervision, S.B.; Writing—original draft, S.B. and R.S.S.; Writing—review and editing, S.B. All authors have read and agreed to the published version of the manuscript.

**Funding:** This research has been performed using the ERIGrid Research Infrastructure and is part of a project that has received funding from the European Union's Horizon 2020 Research and Innovation Programme under Grant Agreement No. 654113. The support of the European Research Infrastructure ERIGrid and its partner PNDC (University of Stratchlyde), Cumbernauld, UK is very much appreciated.

**Acknowledgments:** The authors would like to thank the staff of Power Networks Demonstration Center, Cumbernauld, UK for the technical and administrative support provided during the execution of reported experiment.

**Conflicts of Interest:** The authors declare no conflict of interest. The funders had no role in the design of the study; in the collection, analyses, or interpretation of data; in the writing of the manuscript, or in the decision to publish the results.

## Abbreviations

The following abbreviations are used in this manuscript:

| | |
|---|---|
| PV | Photovoltaic |
| PQ | Power Quality |
| PMU | Phasor Measurement Unit |
| PPS | Pulse Per Second |
| HV | High Voltage |
| RG | Renewable Generation |
| EV | Electric Vehicle |
| MV | Medium Voltage |
| LV | Low Voltage |
| VT | Voltage Transducer |
| CT | Current Transducer |
| MG | Motor-Generator |
| THD | Total Harmonic Distortion |

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
