# Peer review of "Multi-Point Time-Synchronized Waveform Recording for the Analysis of Wide-Area Harmonic Propagation"

_applsci, doi:10.3390/app10113869_

Round 1

Reviewer 1 Report

Dear authors,

I think that your work is very interesting but it should be improved in some ways.

  1. English grammar and wording should be reviewed.
  2. Section 3.3 should be reviewed. For example equation 2 is not correct (k in the exponential should be n). Moreover, I do not understand equation 3 (Is some summation sign missed?). Please clarify the meaning of h and ωh
  3. Please take care of the acronyms you are using. You should define them. What is Fs, sampling frequency? (line 156). What is MG, Motor-generator? (line 171, but first appearance in line 124).
  4. In section 3.4, when you present the test cases, for each configuration (A and B) I miss the case with all the loads disconnected. This is very important to determine background harmonics in order to justify your discussion later. This need is also highlighted at the end of section 4.1 where you had to measure the harmonic current injection of EV charger in order to justify your results. Direct measurement of background harmonics had facilitated all the study and its interpretation.
  5. In the beginning of section 4, you are saying that an upstream network was decoupled from the MV test facility. ¿Which one is it? If it is included in the scheme of Figure 1 you should tell which one is. If it is not included, I do not understand as you are feeding the system with the MG. Please clarify this and its influence in background harmonics. 
  6. In section 4, you are presenting the harmonics in Amps and Volts. This makes difficult comparing between different drawings and even in the text. Please change to percentages. 
  7. Also in section 4 you are presenting all harmonics drawings in hertz (x axis) but in the text you are mixing and sometimes you use the harmonic order. Please unify to make easier the understanding.
  8. As the authors correctly highlight in the conclusions, some calculations can be done to recreate measurements and confirm method correctness. I miss, at least, some basic calculations to reflect and/or confirm experiments. For example system impedance at measurement points.

Author Response

Dear reviewer,

Reviewer 2 Report

- There are a lot of numbers in the paper, but there is no information on the quality of these numbers. Are the measurements traceable? Are they not, but you checked that your instruments are linear with current and voltage? This would allow you to go a long way, since comparisons become possible. Is there a frequency dependency of your instruments? Or do you not rely on the frequency dependency because you don't compare values at different frequencies with each other (such as saying the component at 650 Hz is bigger than that at 150 Hz)? In stating units, you refer to the SI, but without information on the quality of the link between your measurements and the SI, this is meaningless. If all you rely on is the linearity of your voltage and current measurement, individually at each frequency, and the linearity is guaranteed somehow with some quantification (specs?), there is probably no need for a calibration, but please state it clearly. One fundamental principle of science is that findings must be reproducible and comparable. Without a statement on your measurement uncertainty, there cannot be reproducibility and comparability.
This is a fundamental flaw in the research design that must be improved, the only flaw I see. Due to this flaw, it cannot be determined whether the conclusions are supported by the result. Note that this flaw is very fundamental, but probably easy to fix by checking the instruments, their documentation and the set-up, and by adding a short paragraph about this.

Apart from this flaw, the method seems valid, the findings are plausible and a statement of measurement uncertainty would make them relevant.

I have a number of mainly editoral remarks. I would appreciate if more care was taken when writing formulae and mathematical expressions. It is very easy to follow the usual conventions and the reviewers should not be distracted with simple proof reading work. This will also improve their focus and allow them to concentrate on the contents.
Recommended reading: ISO 80000-1 and ISO 80000-2 or BIPM SI brochure (freely available online), NIST SP 811 (freely available online)

- Spacing after full stops is not uniform (e.g. l. 2 vs ll. 4, 95)
- Units are typeset as variables (in italic). The conventional way of typesetting units is in upright (Roman) type. Correct. (e.g. ll. 36, 39)
- There shouldn't be a hyphen between "IEC" and, e.g., "61000". Correct. (e.g. l. 95)
- You are very generous with hyphens (e.g. time-variation, l. 98). Check and remove hyphens if appropriate.
- The PNDC is a proper name. Centre is spelled centre, not center. Correct. (cf. https://www.strath.ac.uk/research/subjects/electronicelectricalengineering/instituteforenergyenvironment/industryengagementresearchcentres/thepowernetworksdemonstrationcentre/ )
- The conventional symbol for the unit ampere is "A", not "Amps". Correct. (l. 130)
- The symbol for the prefix kilo is "k", not "K". "K" means kelvin. Correct. (e.g. Fig. 1)
- Use the approximate sign (ISO 80000-2:2009 item 2-7.5, unicode 0x2248) to say approximately equal, not the proportionality sign (ISO 80000-2:2009 item 2-7.7, unicode 0x223C).
- There is always to be a space between the numerical value and the unit symbol. The only exceptions are the angular unit symbols °, ' and " with are not preceeded by a space. Correct. (e.g. l. 138 "0.5%" -> "0.5 %")
- There are a couple of small issues with the mathematical expressions (they seem minor, but most do change the meaning):
. Often, there is plain text in the indices which is typeset as variables or product of variables (in italic, e.g. (1): index w, (2): STFT = S × T × F × T) and not as text (in upright (Roman)). Correct.
. The variables e, j and pi are not intended the be variables, but constants e (ISO 80000-2:2009 item 2-12.3), j (ISO 80000-2:2009 item 2-14.1) and pi (ISO 80000-2:2009 item 2-13.1), so they must be typeset upright, not italic (e.g. (2)).
. In rare cases, symbols are printed upright in the text (e.g. x[n] in line immediately preceeding (4) -- funny that the line numbers are missing sometimes)
- Line immediately after l. 160: The deviation of the frequency is not delta Hz. It is either delta (with delta = 5 Hz to give a random value) or delta fr. Otherwise, (5) wouldn't work because the dimensions in the parenthesis don't match -- l0 is a dimensionless number (index of the highest component frequency bin) and delta is a frequency). Since in l. 162, you write |delta| <= 0.5, the deviation frequency is probably delta fr as suggested by (5).
- arg is intended to be the well defined function "argument" (ISO 80000-2:2009 item 2-14.5). Typeset as such in upright (Roman) type.
- Don't mix quantities and units (l. 161). A quantity can be expressed in any unit; there is no difference in the quantity length, to give a random example, l = 1000 m and l = 1 km. Remove "in radians".
- There is always to be a space around the equal sign. It's correct in the numbered formulae, but not in the text. Correct. (e.g. l. 162)
- The window using the function named after Mr von Hann is called Hann window, not Hanning window (e.g. l. 165).
- In l. 165, there should be the article "a" in front of Hann. In general, articles are sometimes missing. It's a very minor point, but you might give it a look while you're at it.
- The minus sign must not be used for other purposes such as specifying a range. The specification of ranges must be correct with respect to the units. In l. 184, for instance, the range cannot be from dimensionless 2 to the duration 3 min (and neither 2 (dimensionless) minus 3 min nor 2 min - 3 min = -1 min). It is "2 min to 3 min". Correct.
- I don't understand what to make of the "important remark" starting in l. 199. Especially an "important remark" should be easy to grasp quickly. Is there a current which should not be there in the ideal case, but you know it is there, you can quantify it and remove it's effect on your data completely "its impact on harmonic distortion levels is accounted for" (l. 204)? Then the reader doesn't need to know and the remark, especially with the qualification "important", creates only confusion. Or does this current impact your data? If so, you need to explain the impact more clearly.
- Figures: The lines are hard to see when they coincide with the grid. Can you put markers at the top of the important frequency components? Or can you remove the vertical grid lines? Since the data is mainly a collection of vertical lines, the vertical grid is probably not needed -- contrary to the horizontal grid, which is very helpful.
- Figures: Sometimes, the maximum value on the vertical axis is significantly larger than the maximum value of the data. Wouldn't it be better to optimise the scaling while maintaining the same scaling between related diagrams such as Fig. 2 and Fig. 3?
- Figures: There is plenty of space to the left and right. Can you make the figure wider?
- Figures and surrounding text: In the text, you mix frequencies (e.g. 450 Hz) and harmonic orders. In the figure, there is the frequency. Wouldn't it be easier to read if you used only one way of specifying this information (frequency or harmonic order)? I know the conversion is easy, but the reader should be able to concentrate on the content without getting too distracted by this conversion.
- Figures 4 and 6: The legend is very hard to read. Since there is only one colour in each diagram, you might as well just write what it is in the top part without the colour box.
- Ll. 220ff, ll. 235ff: The difference between Fig. 2 and Fig. 3 could to be due to the EV charger using the grid as part of it's EMC filter. Can you comment on that? Do you know the input circuitry of the EV charger? Is the voltage supplied by the source really the decisive difference? Can you generate a spectrum similar to that of the grid with the MG? Are the results more similar to the other MG results or to the other grid results?
- L. 226 "Figure 2 in Amp scale": It is not an ampere (no capitals -- ampere is the unit, Ampère is the person, A is the conventional unit symbol) scale, but the quantity your are plotting is the current.
- Ll. 224ff etc.: It would help if these differences were visible in a figure. One solution could be to show Fig. 3 next to Fig. 2. You are giving the values of 1.68 V and 0.14 A for one case and say a difference can be noted -- but to not this difference, the reader needs to scroll and take the values for 250 Hz from Fig. 2. This is a huge distraction to the focus of the reader. In addition, 250 Hz coincides with the grid of the plot, so the lines are very hard to see.
- Ll. 249ff: I doubt it is true, but this paragraph looks like all you have found is that you observed an effect related to your set-up that will not appear in a real grid. I think it would be better not to give this impression. Rather than drawing the attention of the reader of the artifacts due to a praticular set-up that the reader doesn't have, you could explain which parts of the findings are relevant to the reader. Perhaps you could grey out the set-up specific data in the plot and add a short statement mentionning that those are set-up specific and not transferable to the real world.
- Figure 5: The style is different for no obvious reason.
- Figure 6: The colour difference on the right hand side is very hard too see in the plots. Some readers might be colour blind, some might not have a colour printer. Please find a different solution.
- Figure 7: This figure allows for a very easy visual comparison of the currents, good! Can this serve as a template for some of the other plots (perhaps without the vertical grid)?
- Figure 7: The colour code is different from that in Figure 6, but the information is related (current and voltage at the same point). Please match the colours.
- Figure 8: The blue dominates the red - the read bars are very hard to make out it the blue is larger. Could a line plot (rather than the bar plot) help disentangle the two data sets?
- Figure 8: The colour of the current axes is not very visible. Could you add the channels to the axis labels ("PV: current magnitude [A]" etc.)?
- Table 2: "A" or "ampere". Writing "Amps" is non-standard (oral use is fine).
- Ll. 338, 407: You are not limited to pure ASCII code, please use the symbol for the unit ohm (capital omega). By the way, Ohm (with capital O) is the family name of the person (and there is only an integer number of persons), the unit name is ohm (see ISO 80000-1:2009 subclause 7.2.5).
- All polar diagrams: The units are missing (V or A and °, I suppose).
- All polar diagrams: It is impossible to see small values of voltage and current since the large values dominate visually. This could be solved easily by plotting points rather than arrows. In Figure 15, for instance, the reader doesn't know yet that you plotted arrows, so I initially thought there value at about 245° was 2.5 V while that at 250° was 3.0 V. As it turns out, the value at 245° is 0 V and all I saw was the tail of the arrow at 250°.
- Table 3: "Emission voltage Sub A" and "RMS" refer to the quantity, "V" to the unit. It is a unusual to put the unit in the middle of the information on the quantity. RMS is not part of the unit.

Author Response

Dear reviewer,

Round 2

Reviewer 1 Report

All my previous comments have been adequately answered.

Author Response

The authors would like to thank this reviewer for the provided feedback and appreciation of our work.

Reviewer 2 Report

The way the authors addressed the comments and the care they took in improving the paper is very good.

I found three minor typos:

Eq. 18: arg{...) should be closed with curly brace.
L. 235: HZ should be Hz.
Eq. 20 and 22: "PCC" and "emission" should be upright.

However, there are two important points which seem to be due to the Latex template and out of the control of the authors. They need to be addressed by the Editorial office.

According to the authors, the Latex template does not allow them to typeset the symbol for microseconds correctly (see l. 36, l.39). The conventional way of typesetting units is in upright (Roman) type (see ISO 80000-1 or BIPM SI brochure (freely available online), NIST SP 811 (freely available online)). This must be fixed.

Some lines are not numbered. This makes references difficult. See, e.g., the entire second half of p. 5. This should be fixed.

Author Response

Thank you for the thorough review of our manuscript. 

We have corrected the remaining typos.

The authors